# Momentum Look-Ahead for Asynchronous Distributed Low-Communication Training

**Thalaiyasingam Ajanthan, Sameera Ramasinghe, Gil Avraham, Yan Zuo, & Alexander Long**
Pluralis Research
`{aj,sameera,gil,yan,alexander}@pluralis.ai`

## Abstract

Distributed Low-Communication (DiLoCo) allows large-scale model training across geographically distributed datacenters by reducing the communication overhead in the data parallel setting. Asynchronous DiLoCo further relaxes the requirement to synchronize the model updates, eliminating any bottlenecks due to slow devices or interconnects. Nevertheless, asynchronous updates introduce *stale (or delayed) gradients* as model updates and gradient computation are no longer synchronized. To alleviate staleness, we introduce a look-ahead based delay correction mechanism by *extrapolating the negative direction of momentum*. Our experiments on language modelling tasks with decoder-only architectures demonstrate that our approach consistently outperforms asynchronous and synchronous DiLoCo methods in both homogeneous and heterogeneous settings.

## 1 Introduction

Data Parallelism (DP) allows large-batch neural network training by replicating the model into multiple devices, and synchronizing the gradients at each iteration with the server to optimize the model parameters (Goyal, 2017). This is communication intensive as the gradients are communicated at each training iteration and the slowest device (or the interconnect) bottlenecks the training due to the synchronization requirement. Inspired by the federated learning literature (McMahan et al., 2017; Reddi et al., 2020), Distributed Low-Communication (DiLoCo) method (Douillard et al., 2023) formulates DP as a bilevel optimization, and shows that gradients can be communicated infrequently without deteriorating convergence, substantially reducing the communication overhead.

The synchronization requirement of DiLoCo is further relaxed in Liu et al. (2024), by allowing each worker (*i.e.*, device) to communicate its gradients to the server independently of other workers. This eliminates the bottleneck due to slow workers, however, introduces asynchronous model updates. Specifically, while some workers are still computing the gradient, the server model would have been updated, leading to *stale (or delayed) gradients*. This presents a significant optimization challenge, necessitating sophisticated delay correction mechanisms, even in the traditional DP setting (Zheng et al., 2017; Stich & Karimireddy, 2019; Assran et al., 2020). Nonetheless, adapting these delay correction mechanisms to DiLoCo requires further investigation (Liu et al., 2024).

In this work, we introduce a look-ahead based delay correction for asynchronous DiLoCo optimization. Our idea is to perform *delay correction in the weight space* by extrapolating the negative direction of momentum. Specifically, we compute momentum as the *exponential moving average of gradients* and by extrapolating the negative momentum direction, our look-ahead step robustly approximates the past gradients steps, alleviating gradient staleness. Our approach is a simple, yet elegant modification to the Nesterov Accelerated Gradient (NAG) method (Nesterov, 1983; 2013) that alters the look-ahead step to act as delay correction in the weight space, without introducing any additional hyperparameters.

We demonstrate the merits of our approach on large-scale language modelling tasks with decoder-only transformer architectures (Vaswani, 2017; Karpathy, 2022) on DiLoCo training with homogeneous and heterogenous devices. In short, our approach consistently outperforms existing asynchronous DiLoCo methods, even surpassing the synchronous baseline.

## 2 PRELIMINARIES

We briefly review the Nesterov Accelerated Gradient (NAG) (Nesterov, 1983; 2013) method and the Distributed Low-Communication (DiLoCo) method (Douillard et al., 2023) upon which we build our work. We refer the interested reader to the respective papers for more details.

### 2.1 NESTEROV ACCELERATED GRADIENT

Nesterov Accelerated Gradient (NAG) (Nesterov, 1983; 2013; Bubeck et al., 2015) is an accelerated gradient method that has the optimal superlinear convergence rate for smooth convex functions in the non-stochastic setting. The main idea is performing a look-ahead step in the previous update direction, combined with a carefully selected step-size sequence to ensure accelerated convergence.

Let $f : \mathbb{R}^m \to \mathbb{R}$ be the objective function. Then, NAG performs the following iterations starting from an initial point $\mathbf{w}_1 \in \mathbb{R}^m$:

$$\mathbf{d}_t = \gamma_t(\mathbf{w}_t - \mathbf{w}_{t-1}) , \tag{1}$$
$$\mathbf{w}_{t+1} = \mathbf{w}_t + \mathbf{d}_t - \eta \nabla f(\mathbf{w}_t + \mathbf{d}_t) .$$

Here, the learning rate $\eta > 0$, the momentum coefficient $\gamma_t$ satisfies, $\gamma_1 = 0$, $0 < \gamma_t < 1$, and the sequence of $\gamma_t$ is derived as part of the convergence proof (Bubeck et al., 2015). Note that, $\mathbf{d}_t$ corresponds to the look-ahead step which extrapolates the update $(\mathbf{w}_t - \mathbf{w}_{t-1})$ by $\gamma_t$ and the gradients are computed at the extrapolated point $(\mathbf{w}_t + \mathbf{d}_t)$.

Analogous to the classical momentum (Polyak, 1964; Sutskever et al., 2013), by denoting the update as $\mathbf{w}_{t+1} = \mathbf{w}_t - \eta \, \mathbf{m}_{t+1}$, the look-ahead can be thought of as taking a step in the negative direction of momentum, *i.e.*, $\mathbf{d}_t = -\eta \, \gamma_t \, \mathbf{m}_t$[1]. Here, the momentum $\mathbf{m}_t$ takes the following form:

$$\mathbf{m}_{t+1} = \gamma_t \, \mathbf{m}_t + \nabla f(\mathbf{w}_t + \mathbf{d}_t) . \tag{2}$$

This interpretation is useful in deriving our momentum based look-ahead method.

NAG has been incorporated into popular deep learning optimizers such as SGD (Sutskever et al., 2013) and Adam (Dozat, 2016), although it often slightly underperforms for usual neural network training. Nevertheless, for DiLoCo setting, it has shown to be superior in both synchronous and asynchronous outer-optimization (Douillard et al., 2023; Liu et al., 2024), and we further improve it by adapting the look-ahead step.

### 2.2 DISTRIBUTED LOW-COMMUNICATION TRAINING

In a traditional DP setup (Goyal, 2017), the dataset is split into multiple worker nodes where each worker computes the gradients on its data-split, and communicates it to the parameter server. The server then updates the parameters by aggregating the gradients from all workers and distributes the updated parameters back to the workers for the next iteration. This process is communication intensive as the gradients are communicated at each iteration.

Inspired by the federated learning literature (McMahan et al., 2017; Reddi et al., 2020), DiLoCo (Douillard et al., 2023) shows that infrequent communication with the server is sufficient for convergence. Specifically, DiLoCo formulates DP as a bilevel optimization problem, where the inner-optimization is performed on each worker (*i.e.*, local SGD) and the server model is updated by synchronizing the model parameters from all workers (*i.e.*, outer-optimization), at a predefined interval (*e.g.*, every 50 inner-iterations).

Formally, let $\mathbf{w}_t^i$ be the local model parameters of worker $i$ at time step $t$. It performs multiple optimization steps locally using its own data-split to obtain $\mathbf{w}_{t+1}^i$, and communicates the weight difference $\mathbf{g}_t^i := \mathbf{w}_t^i - \mathbf{w}_{t+1}^i$ (*i.e.*, pseudogradient)[2] to the server. The server model receives the pseudo-gradients $\{\mathbf{g}_i\}$ from all workers, and updates its parameters in the negative direction of the average pseudogradient. Precisely, let $\mathbf{w}_t$ be the server model parameters, then the outer-optimization step can be written as:

---

[1]This is simply a rewrite of $\mathbf{d}_t$ in Eq. (1) in terms of $\mathbf{m}_t$.

[2]This approximates the gradient of the inner-optimization process with respect to the model parameters using first-order difference.

$$\mathbf{w}_{t+1} = \mathbf{w}_t - \eta \frac{1}{k} \sum_{i=1}^{k} \underbrace{\overbrace{\left[\mathbf{w}_t^i - \mathbf{w}_{t+1}^i\right]}^{\mathbf{g}_t^i}}_{\mathbf{g}_t}, \tag{3}$$

where $\eta > 0$ is the learning rate, and $k$ is the number of workers. We omit optimizer specific updates for brevity. Then, the server communicates the updated parameters to all workers, setting their parameters as $\mathbf{w}_{t+1}^i = \mathbf{w}_{t+1}$ for all $i$, for the next iteration. In DiLoCo, since all workers are synchronized at each outer optimization step $t$, $\mathbf{w}_t^i = \mathbf{w}_t$ for all $i$, hence, the update in Eq. (3) takes a step towards the average of all worker model parameters. By taking a small step rather than directly averaging worker model parameters, DiLoCo stabilizes training.

**Asynchronous DiLoCo.** The synchronization requirement in DiLoCo is a latency bottleneck due to the communication overhead and/or heterogeneity of devices. Relaxing this requirement alleviates the bottleneck at the cost of *gradient staleness*, which we discuss below.

In asynchronous DiLoCo (Liu et al., 2024) each worker communicates its parameters to the server independently to other workers, leading to asynchronous updates of the server model. Precisely, the outer-update takes the following form:
$$\mathbf{w}_{t+1} = \mathbf{w}_t - \eta \left[\mathbf{w}_t^i - \mathbf{w}_{t+1}^i\right] . \tag{4}$$

Note, the server updates its parameters whenever it receives the pseudogradient from a worker[3], and therefore $\mathbf{w}_t$ would have been updated multiple times, while worker $i$ is performing its inner-optimization. Therefore, $\mathbf{w}_t^i$ is *older* compared to the server parameters $\mathbf{w}_t$, and consequently, the pseudogradients are also stale. Suppose the delay for worker $i$ be $\tau_i$, then $\mathbf{w}_t^i = \mathbf{w}_{t-\tau_i}$. Substituting this in Eq. (4):
$$\mathbf{w}_{t+1} = \mathbf{w}_t - \eta \underbrace{\left[\mathbf{w}_{t-\tau_i} - \mathbf{w}_{t+1}^i\right]}_{\bar{\mathbf{g}}_t^i} . \tag{5}$$

The *stale (or delayed) pseudogradient* is denoted as $\bar{\mathbf{g}}_t^i$. This gradient staleness causes optimization challenges and sophisticated delay correction mechanisms need to be employed.

In Liu et al. (2024), a buffer-based Nesterov approach is employed to aggregate the pseudogradients so as to stabilize the outer-optimization. Specifically, pseudogradients are accumulated into a fixed size buffer, and when the buffer is full, Nesterov method is applied. For all other iterations (*i.e.*, when the buffer is being filled) standard SGD is employed. Additionally, the number of inner-optimization steps are adjusted to cater for the heterogeneity of devices. We will show subsequently that, the look-ahead step in the Nesterov method can be repurposed to handle staleness in pseudogradients without any additional heuristics or hyperparameters.

## 3 METHOD

We follow the asynchronous DiLoCo (Liu et al., 2024) setup and derive a momentum based look-ahead method inspired by NAG to mitigate gradient staleness in the outer-optimization. Our method extrapolates the momentum direction, so that the pseudogradients are computed at a point that is closer to the ideal one. This is appealing as it does not make any assumptions about the loss function or gradients as in the delay correction methods (Zheng et al., 2017; Xie et al., 2019) tested in Liu et al. (2024). The only assumption is that the outer-update directions can be approximated using the momentum, which is valid for momentum-based optimizers.

### 3.1 MOMENTUM LOOK-AHEAD FOR DELAYED GRADIENTS

Let us consider a particular worker and drop the worker index for brevity. Suppose $f : \mathbb{R}^m \to \mathbb{R}$ be the function for which the pseudogradient $\nabla f$ is computed. Now, our momentum look-ahead method can be written as:

$$\begin{aligned}
\mathbf{d}_t &= -\eta \, \gamma_t \, \mathbf{m}_t , & \text{look-ahead} && (6) \\
\mathbf{w}_{t+1} &= \mathbf{w}_t + \mathbf{d}_t - \eta \nabla f(\bar{\mathbf{w}}_t + \mathbf{d}_t) , & \text{weight update} \\
\mathbf{m}_{t+1} &= \gamma_t \, \mathbf{m}_t + (1 - \gamma_t) \nabla f(\bar{\mathbf{w}}_t + \mathbf{d}_t) . & \text{momentum computation}
\end{aligned}$$

---

[3]Asynchronous DiLoCo performs more frequent outer-updates (up to $k\times$ more) compared to DiLoCo.

Here, $\bar{\mathbf{w}}_t$ denotes the delayed point, $\bar{\mathbf{w}}_t = \mathbf{w}_{t-\tau} = \mathbf{w}_t - \Delta_t$, where $\tau$ is the delay. The main difference compared to NAG is the momentum computation, where we discount the gradient term by $(1 - \gamma_t)$ (in contrast to Eq. (2)). This ensures that the momentum is an exponential moving average of gradients. Intuitively, momentum in our case is a smooth approximation of gradients, and look-ahead in the negative direction of momentum robustly approximates past gradient steps.

Specifically, due to the modification above, the look-ahead direction $\mathbf{d}_t$ does not extrapolate the previous update direction $(\mathbf{w}_t - \mathbf{w}_{t-1})$ as in NAG, instead it follows the negative momentum direction. Since the pseudogradient $\bar{\mathbf{g}}_t := \nabla f(\bar{\mathbf{w}}_t + \mathbf{d}_t)$ is noisy due to staleness, the update direction can be noisy, however, momentum being the moving average of gradients, provides a robust look-ahead direction. Alternatively, $\mathbf{m}_t$ can be thought of as a proxy for latest gradient information as it is up to date (*i.e.*, no staleness) and doing a look-ahead in the direction reduces staleness. This subtle but important difference, ensures that our update is an effective delay correction mechanism ensuring empirical convergence in the presence of gradient staleness.

Note, in deep learning, $\nabla f$ is computed using automatic differentiation. Therefore, when NAG is incorporated into deep learning optimizers (Sutskever et al., 2013; Dozat, 2016), $\boldsymbol{\theta}_t := \mathbf{w}_t + \mathbf{d}_t$ reparametrization is used to simplify the implementation. To this end, our reparametrized updates can be written as:

$$\mathbf{m}_{t+1} = \gamma_t \, \mathbf{m}_t + (1 - \gamma_t) \nabla f(\bar{\boldsymbol{\theta}}_t) \,, \qquad \text{momentum computation} \qquad (7)$$
$$\boldsymbol{\theta}_{t+1} = \boldsymbol{\theta}_t - \eta \left[ \gamma_{t+1} \, \mathbf{m}_{t+1} + \nabla f(\bar{\boldsymbol{\theta}}_t) \right] \,, \qquad \text{weight update}$$

where $\bar{\boldsymbol{\theta}}_t := \bar{\mathbf{w}}_t + \mathbf{d}_t$. These updates constitute to a *one-line change* in the SGD implementation and can be easily incorporated. The pseudocode of our algorithm is provided in Appendix A.

Conceptually, the buffer-based approach presented in Liu et al. (2024) can be interpreted as a special case of our method. In particular, the exponential moving average aggregates the gradients online as opposed to a fixed size buffer, and enables us to apply Nesterov updates at every step, eliminating the need for any additional heuristics or hyperparameters. As shown in our experiments, our method consistently outperforms the buffer-based gradient aggregation. Furthermore, adjusting the number of inner-optimization steps to cater for heterogeneous devices improves all asynchronous methods, including ours.

## 4 RELATED WORK

**Asynchronous data parallel methods.** DP is a traditional distributed training setting, where each device optimizes the full model and periodically synchronizes the model parameters. Asynchronous DP methods are well-studied within the theoretical framework and many gradient delay correction mechanisms have been developed (Agarwal & Duchi, 2011; Stich & Karimireddy, 2019; Assran et al., 2020). Notable methods that improve over the simple asynchronous SGD (Recht et al., 2011) include delay dependent learning rate (Barkai et al., 2019; Mishchenko et al., 2022), gradient forecasting with second-order information (Zheng et al., 2017), and look-ahead in the weight space (Hakimi et al., 2019). Some of these approaches (Zheng et al., 2017; Xie et al., 2019) are shown to underperform in the asynchronous DiLoCo setting (Liu et al., 2024), and therefore, further investigation is required when adopting them to this setting. Apart from this, training dynamics of asynchronous DP methods have also been analyzed (Mitliagkas et al., 2016) and some of these observations may be useful in the DiLoCo setting as well.

**DiLoCo methods.** DiLoCo improves the communication requirement of DP by showing that infrequent synchronization of server model parameters is sufficient for convergence. This setup is inspired by the federated learning literature, where this local-SGD (*i.e.*, inner-optimization of DiLoCo) is well-studied (McMahan et al., 2017; Reddi et al., 2020; Yang et al., 2022). Recent works, further improve the efficiency of DiLoCo with asynchronous outer-optimization (Liu et al., 2024) or via clever methods to mask the communication overhead (Douillard et al., 2025). In this work, we consider asynchronous DiLoCo setup and introduce a variant of NAG for the outer-optimization to mitigate gradient staleness.

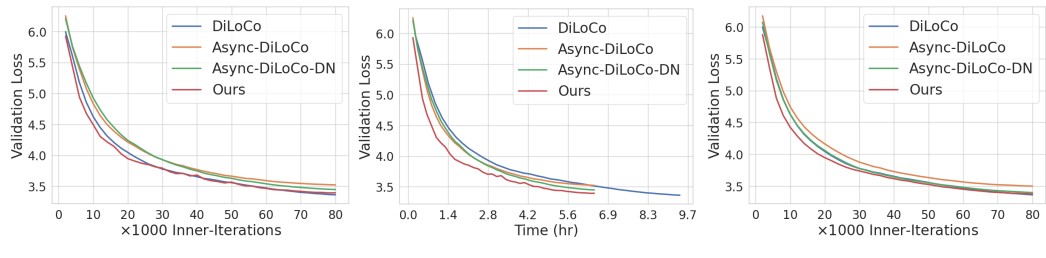

(a) Heterogeneous, w.r.t. iterations    (b) Heterogeneous, w.r.t. time    (c) Homogeneous

Figure 2: *Results on WikiText with* 4 *workers. For both homogenous and heterogeneous settings, our method clearly outperforms the asynchronous methods while being competitive to the synchronous DiLoCo. Furthermore, asynchronous methods are much faster in the heterogeneous setup.*

## 5 EXPERIMENTS

### 5.1 RESULTS ON A TOY DATASET

As a sanity check, we first test our method on a toy dataset provided by Liu et al. (2024). This is a classification task on a mixture of mixtures on Gaussian data using a Multi-Layer Perceptron (MLP). This is proposed as a minimal example that replicates the behaviour of large-scale asynchronous DiLoCo. We use the provided code[4] and simply plug-in our optimizer while keeping everything else unchanged. All methods use SGD+Nesterov as the outer optimizer. As per the code, we use learning rate of 0.7 for all methods except for Async-DiLoCo which uses 0.07 and the number of workers is set to 4. Async-DiLoCo-DN is the delayed Nesterov method proposed in (Liu et al., 2024) which we implement based on the provided pseudocode and the buffer size is set to 4. For this experiment, for our method, $\gamma_t$ in Eq. (6) is set to 0.99.

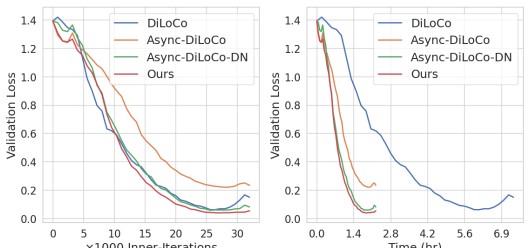

Figure 1: *Results on a toy classification dataset. Our method clearly outperforms both asynchronous and synchronous methods.*

As shown in Fig. 1, Async-DiLoCo-DN outperforms Async-DiLoCo and the synchronous DiLoCo method, and our method improves over Async-DiLoCo-DN. This experiment validates our implementation of Async-DiLoCo-DN and demonstrates the merits of our momentum look-ahead method.

### 5.2 LANGUAGE MODEL EXPERIMENTS

We evaluate our method on the WikiText (Merity et al., 2016) using decoder-only architectures. Our model architecture is based on NanoGPT (Karpathy, 2022) with no dropout. The base configuration includes a context length of 512, an embedding dimension of 768, 12 attention heads, and 6 layers, with approximately 90M parameters. We use the `Distilled-BERT-Uncased` tokenizer (Sanh et al., 2019) and *train the model from scratch*. Across all experiments, we maintain a batch size of 8, a inner-optimization learning rate of $3e$-4, and a weight decay of 0.1, unless otherwise specified. AdamW (Loshchilov, 2017) is used as the inner-optimizer with with a linear warmup of 1k iterations starting from a learning rate of $1e$-7. The learning rate is then decayed to $3e$-5 following a cosine decay schedule. The number of inner-optimization steps is set to 50 and the experiments are run for a total of 80k inner-steps.

We use SGD+Nesterov as the outer optimizer for all methods. For the 4-worker experiment, a constant learning rate of 0.7 is set for all the methods, except Async-DiLoCo which uses 0.07 as it was unstable with larger learning rate. The buffer size of Async-DiLoCo-DN is set to 4 for all experiments. Default values of momentum is used for all methods including ours where $\gamma_t$ is set to 0.9. All experiments are run on a system equipped with 8 A10G GPUs. The heterogeneous setup is simulated by sampling workers from different device speeds: $\{1.70, 1.33, 0.66, 0.30\}$, where each entry corresponds to time taken per inner-iteration in seconds. The results are reported in Fig. 2.

---

[4]https://github.com/google-deepmind/asyncdiloco

Our method outperforms the asynchronous methods while being competitive to the synchronous DiLoCo in both heterogenous and homogenous setups. Furthermore, asynchronous methods are much faster in the heterogeneous setup. Notably, our method which extrapolates in the momentum direction, is significantly better in the initial phase of training where the step sizes are larger and noisier, validating our insight.

**Dynamic local updates.**

Liu et al. (2024) discusses an approach to dynamically adjust the number of inner-optimization steps to cater for heterogeneous devices, where the slower device executes proportionally fewer number of steps, balancing the time taken per inner-optimization in each worker. Please refer to Eq. (6) of Liu et al. (2024) for the exact formula. This is called, Dynamic Local Updates (DyLU), which brings the heterogenous setup closer to a homogeneous one and improves all asynchronous methods, including ours.

We test DyLU on the WikiText dataset with 4 workers. As reported in Fig. 3, with DyLU, all asynchronous methods outperform the synchronous DiLoCo, where our method yields the

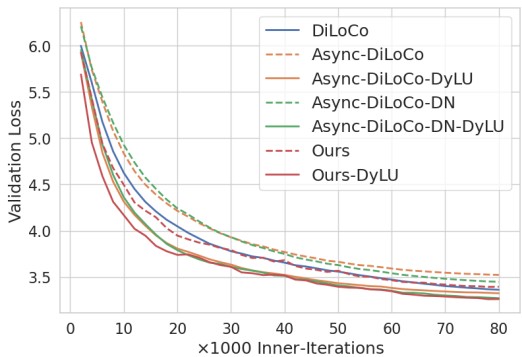

Figure 3: *Dynamic local updates improves all asynchronous methods. With DyLU, asynchronous methods clearly outperform the synchronous DiLoCo method.*

best performance. It is intriguing to observe that asynchronous methods can outperform synchronous methods in DiLoCo. One plausible explanation is that it may be due to asynchronous methods performing more frequent outer-updates (see Sec. 2.2) for the same amount of data (*i.e.*, total inner-optimization steps). Nevertheless, this warrants further investigation.

**Increasing the number of workers.**

To test the robustness, we increase the number of workers to 8 and evaluate the methods. We simulate the heterogenous setup as described above by sampling device speeds. We ran all the methods for 80k inner-iterations, with a constant learning rate of 0.5 for all methods, except for Async-DiLoCo which uses 0.07. All other hyperparameters remain unchanged. This experiment was performed on a system equipped with 8 A100 GPUs.

As reported in Fig. 4, the results are consistent

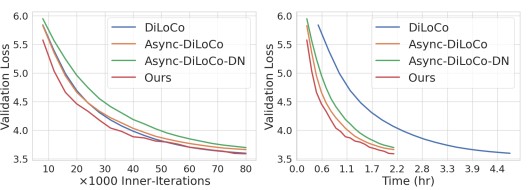

Figure 4: *Results on WikiText with 8 workers on a heterogeneous setup. Consistent with 4 workers experiment, our method outperforms asynchronous methods while being competitive with synchronous DiLoCo.*

with the 4 worker setup. In short, our method outperformed the asynchronous methods and it is competitive with synchronous DiLoCo. Our method is significantly better for the first half of training but the synchronous DiLoCo bridges this gap towards the end of training. Tuning the learning rate schedule for the outer-optimization might improve our method, but we leave such explorations for future work. Interestingly, Async-DiLoCo is better than Async-DiLoCo-DN for most part of training. Although intriguing, it indicates that more frequent updates using Nesterov method is beneficial, however, in-depth investigation is required.

## 6    CONCLUSION

We introduce a simple momentum based look-ahead method to alleviate gradient staleness in asynchronous DiLoCo optimization. Our method is a one-line change in the SGD+Nesterov optimizer and does not introduce any additional hyperparameters. As shown in our experiments, it consistently outperforms synchronous and asynchronous DiLoCo methods in both homogeneous and heterogeneous settings. In future, we intend to theoretically analyze the convergence properties of our method in the presence of stale gradients.

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

## A  APPENDIX

Below, we provide the pseudocode of our algorithm. For specific implementation details of the server and worker logic, we refer the reader to the code provided by (Liu et al., 2024)[5].

---

**Algorithm 1** Momentum Look-Ahead for Asynchronous DiLoCo

---

**Require:** $K$ replicas, $H$ inner-steps, $T$ total inner-steps, loss function $f$, data shards $\{\mathcal{D}_1, \ldots, \mathcal{D}_K\}$
**Require:** Initial model $\boldsymbol{\theta}_0$, learning rate $\eta$, momentum coefficient $\gamma$, optimizer `InnerOpt`

      **At each worker** $k = 1 \ldots K$ **(in parallel):**            ▷ Worker logic as in (Liu et al., 2024)
1: Initialize local model: $\boldsymbol{\theta}_0^k \leftarrow \boldsymbol{\theta}_0$
2: **for** $t = 1 \ldots T$ **do**
3:      $\mathbf{x} \sim \mathcal{D}_k$            ▷ Sample local data
4:      $\mathcal{L}_t^k \leftarrow f(\mathbf{x}, \boldsymbol{\theta}_{t-1}^k)$            ▷ Compute loss
5:      $\boldsymbol{\theta}_t^k \leftarrow \texttt{InnerOpt}(\boldsymbol{\theta}_{t-1}^k, \nabla \mathcal{L}_t^k)$            ▷ Local update
6:      **if** $t \bmod H = 0$ **then**
7:          $\boldsymbol{\Delta}_t^k \leftarrow \boldsymbol{\theta}_{t-H}^k - \boldsymbol{\theta}_t^k$            ▷ Compute pseudogradient
8:          $\texttt{send}(\boldsymbol{\Delta}_t^k)$ to server            ▷ Send pseudogradient
9:          $\boldsymbol{\theta}_{t+1}^k \leftarrow \texttt{recv}(\boldsymbol{\theta}_{t+1})$            ▷ Pull updated model from server
10:      **end if**
11: **end for**

      **At server (runs continuously):**            ▷ Server logic as in (Liu et al., 2024)
12: **while** True **do**
13:      $\boldsymbol{\Delta}_t \leftarrow \texttt{recv}(\boldsymbol{\Delta}_t^k)$            ▷ Receive pseudogradient
14:      $\boldsymbol{\theta}_{t+1} \leftarrow \texttt{OuterOpt}(\boldsymbol{\theta}_t, \boldsymbol{\Delta}_t)$            ▷ Update server model
15:      $\texttt{send}(\boldsymbol{\theta}_{t+1})$ to worker $k$            ▷ Send updated model
16: **end while**

17: Initialize optimizer state: $\mathbf{m}_0 \leftarrow \mathbf{0}$
18: **function** OUTEROPT($\boldsymbol{\theta}_t, \boldsymbol{\Delta}_t$)            ▷ Momentum look-ahead as in Eq. (7)
19:      $\mathbf{m}_{t+1} \leftarrow \gamma \, \mathbf{m}_t + (1 - \gamma) \, \boldsymbol{\Delta}_t$            ▷ Momentum update
20:      $\boldsymbol{\theta}_{t+1} \leftarrow \boldsymbol{\theta}_t - \eta \, (\gamma \, \mathbf{m}_{t+1} + \boldsymbol{\Delta}_t)$            ▷ Weight update
21:      **return** $\boldsymbol{\theta}_{t+1}$
22: **end function**

---

[5]https://github.com/google-deepmind/asyncdiloco

