# OpenReview forum: "Momentum Look-Ahead for Asynchronous Distributed Low-Communication Training"
_ICLR.cc/2025/Workshop/MCDC — MCDC @ ICLR 2025_

### Official Review · Reviewer_XQpt · 2025-02-26

**Rating:** 7
**Confidence:** 3
**Fit:** 4

**Summary:**

This work provides an extension to Asynchronous DiLoCo taking inspiration from Nesterov Accelerated Gradient.

By taking a 'look-ahead' step in the direction of negative momentum, convergence can be accelerated as losses from out-of-sync updates (Async DiLoCo) are minimized.

The paper demonstrates experimentally the strength of DiLoCo+NAG, showing that their technique can even outperform vanilla DiLoCo in both number of iterations to convergence and wall-clock time to convergence.

**Reason For Giving A Higher Score:**

Strong experimental results with a clear mathematical foundation and motivation.

**Reason For Giving A Lower Score:**

Not enough discussion of *why* the results are so strong.

**Strengths And Weaknesses:**

The mathematics of this paper is strongly defined and well-cited.

The experimental results are persuasive but I felt a lack of explanation of why this method has been shown to be *so* powerful compared with previous works.

**Suggestions:**

More explanation of why this method is so strong would be beneficial; instead, I found discussion of the specifics of implementation (eq 7) which were less necessary.

I was confused by Eq. 1, as they define the outer-optimization step of DiLoCo as pure SGD - there is no reference to the momentum or other optimizer terms. I think this is just a typo.

---

### Official Review · Reviewer_542C · 2025-02-27

**Rating:** 8
**Confidence:** 3
**Fit:** 5

**Summary:**

In this paper, authors propose a look-ahead based delay correction to reduce the effect of stale gradients for Async Diloco training. They perform delay corrections by extrapolating the negative direction of momentum. In this way, they take into account the previous gradients steps and avoid staleness.

It is a modification of Nesterov acc. grad. method where in the momentum computation the gradient term is multiplied by a factor (1-\gamma_j).

The authors tested their modification with a toy dataset over a simple MLP, and WikiText over NanoGPT (90M) architecture. In both cases, they achieve better results wrt. Async-Diloco and compatible results wrt. sync. Diloco.

**Reason For Giving A Higher Score:**

- simple yet effective method improving performance in the async setting

**Reason For Giving A Lower Score:**

- limited experiments (regarding model and datasets)

**Strengths And Weaknesses:**

Strengths:

- proposed a modified Nesterov method that performs better than existing ones
- well structured paper

Weaknesses:

- limited novelty

**Suggestions:**

- further experimenting with different datasets and model architectures

---

### Official Review · Reviewer_UTA3 · 2025-02-28

**Rating:** 4
**Confidence:** 2
**Fit:** 5

**Summary:**

The paper proposes a Momentum Look-Ahead mechanism to address gradient staleness in asynchronous Distributed Low-Communication (DiLoCo) training, adapting Nesterov Accelerated Gradient (NAG) by using an exponential moving average (EMA) of gradients as momentum to correct delays. Experiments on a toy dataset and WikiText language modeling with a 90M-parameter decoder-only transformer show the method outperforms synchronous and asynchronous DiLoCo. However, the contribution’s clarity and theoretical grounding remain unclear to me.

**Reason For Giving A Higher Score:**

- Empirical Results: Strong performance on WikiText (Figure 2) suggests practical utility, warranting further exploration.

**Reason For Giving A Lower Score:**

- Lack of Clarity: Inconsistencies in NAG formulation, poor section ordering, and unclear intuition for EMA vs. NAG momentum make the method hard to understand, especially for someone less familiar with this area.
- Experimental Concerns: The potential regularization effect of async noise (Figures 3, 4) raises doubts about the setup’s validity, undermining confidence in the results.

**Strengths And Weaknesses:**

Strengths:
- Empirical Success: Figure 2 demonstrates improved performance over previous DiLoCo methods on WikiText, suggesting practical value.

Weaknesses:
- Order and Clarity of NAG Discussion: The buffer-based Nesterov approach (Liu et al., 2024) is introduced before NAG in Section 2, disrupting the flow and making it hard to follow the progression to Section 3.1, where the proposed method lacks sufficient detail for me to grasp its implementation.
- Inconsistent NAG Definition: Lines 120–131 describe NAG without scaling the look-ahead d_t by the learning rate, but later (lines 134, Eq 5 and 7) d_t is scaled by learning rate, deviating from typical NAG (e.g., Sutskever et al., 2013). This inconsistency confuses the method’s alignment with classical NAG.
- Experimental Concerns: Figures 3 and 4 suggest asynchronous methods outperform synchronous DiLoCo for the same number of iterations, raising concerns about regularization via noise in async setups potentially speeding optimization unnaturally, which could skew results.
- Unclear Intuition for EMA vs. Regular NAG: Replacing NAG’s momentum with an EMA of gradients isn’t intuitively justified. The paper doesn’t clarify why EMA’s properties (e.g., smoothing) are better than regular NAG momentum in asynchronous DiLoCo, especially given my limited understanding of Liu et al.’s async setup.

**Suggestions:**

- Address NAG Inconsistency: Explain why learning rate is included in the look-ahead step
- Validate Experimental Setup: Investigate whether async noise acts as regularization, potentially explaining faster convergence. Compare training steps across sync and async setups to ensure fairness, and discuss noise effects explicitly.
- Reorder and Clarify Sections: Move the NAG background before discussing the buffer-based approach to establish a clear foundation. Expand Section 3.1 with detailed pseudocode or figure of the async DiLoCo setup

---

### Decision · Program_Chairs · 2025-03-06

**Decision:**

Accept

**Comment:**

This work has been praised by some reviewers that recognized its merits, but presentation and clarity could be improved. We recommend the authors to take into consideration reviewer UTA3's comments in their final version.